**Data Availability Statement:** Data will be provided as supporting information files upon acceptance of the manuscript.

**Funding:** The authors received no specific funding for this work.

# Asymmetric changes in foot anthropometry with pregnancy may be related to onset of lower limb and low back pain

Erica M. Casto[1☯¤a], Corrie Mancinelli[2‡], Petronela Meszaros[3¤b‡], Jean L. McCrory[1☯]*

1 Division of Exercise Physiology, Department of Human Performance and Applied Exercise Science, West Virginia University School of Medicine, Morgantown, West Virginia, United States of America, 2 Division of Physical Therapy, Department of Human Performance and Applied Exercise Science, West Virginia University School of Medicine, Morgantown, West Virginia, United States of America, 3 Department of Obstetrics and Gynecology, West Virginia University School of Medicine, Morgantown, West Virginia, United States of America

☯ These authors contributed equally to this work.
¤a Current address: Department of Kinesiology, University of Massachusetts, Amherst, Massachusetts, United States of America
¤b Current address: Department of Obstetrics and Gynecology, Atrium Health, Charlotte, North Carolina, United States of America
‡ CM and PM also contributed equally to this work.
* jlmccrory@hsc.wvu.edu

## Abstract

### Introduction

Fifty percent of pregnant females experience pain with 20% reporting long-term pain post-partum. Pregnant females undergo changes in foot anthropometry, lower extremity alignment, and joint laxity. It is unknown if asymmetric alterations may be related to development of pain. The **purpose** of this study was twofold: 1) to compare asymmetric alignment in pregnant females with and without pain during pregnancy and in nulliparous controls and 2) to assess the relationship between asymmetric alignment and pain severity in all participants.

### Methods

Ten pregnant females in their third trimester and nine nulliparous controls participated. Bilateral asymmetry of foot length, width, arch index, arch height index, arch rigidity index, arch drop, rearfoot angle, and pelvic obliquity were determined. Joint laxity and musculoskeletal pain were also assessed. ANOVAs were utilized to compare asymmetries between pregnant females reporting pain (n = 5), those not reporting pain (n = 5), and controls. Spearman's Rho correlations were used to relate asymmetry to pain magnitude (α = 0.05).

### Results

No statistical differences (p>0.05) were found between pregnant females with or without pain and controls for any of the metrics. Negative correlations were found between arch index asymmetry and low back pain (p = 0.005), foot length asymmetry and lower leg pain

**Competing interests:** The authors have declared that no competing interests exist.

(p = 0.008), and pelvic obliquity and lower leg pain (p = 0.020). Positive correlations were found between foot width asymmetry and knee pain (p = 0.028), as well as arch drop asymmetry and upper leg (p = 0.024), knee (p = 0.005), and lower leg pain (p = 0.019).

## Conclusions

This study was successful in identifying potential targets for prevention and treatment of pain in pregnancy. Furthermore, because pain during pregnancy may be predictive of pain post-partum, it is important to conduct future research to determine both if interventions such as footwear or exercise can prevent or treat these asymmetries and prevent post-partum pain.

## Introduction

Approximately 50–75% of pregnant females report low back and lower extremity pain during pregnancy, with about a quarter reporting severe pain reducing overall quality of life [1–7]. Furthermore, 20% of females report continued pain post-partum [1, 4–6, 8]. This pain may be attributed to the anatomic and physiological changes that occur throughout pregnancy [5]. Among these changes are, most notably, increased abdominal volume, altered thoracopelvic alignment, and increased joint laxity [9–12]. Relaxin, which peaks in the first trimester of pregnancy, increases ligamentous laxity to allow for pelvic girdle expansion; however, it has been reported to affect peripheral joints as well [5, 9, 13–15]. Moreover, prior work has shown a relationship between joint laxity and the occurrence of low back pain during pregnancy and post-partum [14].

Foot length and width both have been shown to increase during pregnancy [5, 16–19] which likely result from increased ligamentous laxity. Foot length, specifically, has been shown to endure post-partum [18, 19]. Additionally, Segal et al. [19] reported increased arch drop, and reduced arch height and rigidity during pregnancy; however, there is limited quantification of how these specific changes in width or length correlate to pain. Damen et al. [20] reported an association between asymmetric laxity and pelvic pain during pregnancy, but it is not clear if this could be related to asymmetric structural alignment or to asymmetric laxity as only laxity was studied. Harrison [18] performed repeated measures of static foot and lower limb alignment during each trimester of pregnancy and post-partum and reported a relationship between arch height and flexibility metrics and reports of pain. However, most participants in the study reported low levels of pain, so true correlations of lower extremity and foot alignment changes with pain could not be assessed. Thus, the relationship of structural asymmetries in the lower extremity and pain during pregnancy is not clear and investigation of the role alignment asymmetries and laxity may play in development of pain during pregnancy is needed. Our study had two primary aims: 1) to quantify differences in asymmetrical lower limb alignment, defined as the absolute difference between right and left sides, as well as ligamentous laxity between pregnant females who report pain, pregnant females who do not report pain, and nulliparous controls and 2) to examine the relationship between asymmetry and pain in all subjects. We hypothesized that 1) pregnant females who reported pain would have increased lower extremity and foot asymmetries and increased overall laxity in comparison to pregnant females who did not report pain as well as the non-pregnant controls, and 2) that the magnitude of biomechanical asymmetry in the pelvis and foot would be positively related to the severity of pain. It is expected that these results may lead to development of

strategies to limit the occurrence or severity of such pain, either by orthotic and footwear interventions or targeted rehabilitation and exercise interventions. Due to the prevalence of long-term pain postpartum [1, 4–6, 8], there is significant need for conservative treatment and prevention strategies of such pain.

## Methods

### Participants

Nineteen healthy females ages 21–34 participated in this study including ten pregnant females and nine nulliparous controls. Participants were recruited from the university physicians' obstetrics practice as well as through word-of-mouth from the local community. Females reporting chronic pain or with known diagnosed scoliosis were excluded in order to avoid confounding factors contributing to these asymmetries or pain [21]. Participants were also excluded if they reported a history of lower extremity injury, fractures or surgeries within a year, diabetes, smoking, or medical conditions affecting sensation or contributing to pain levels [10, 22]. Demographic data for the pregnant and control participants are shown in Table 1. Pregnant females reported a greater mass prior to pregnancy than the current mass of the non-pregnant females (p = 0.001), but no differences in age and height were noted between groups. Females in the pregnant group were subcategorized into either the "pregnant pain" group or "pregnant no pain" group based on their responses to the VAS pain questionnaires. This will be explained in further detail in the procedures section below. There were no differences in age, height, current mass, pre-pregnancy mass, weeks pregnant, and number of pregnancies between the pregnant pain and pregnant no-pain groups. Demographic data are provided in Table 2.

### Procedures

Following an explanation of study procedures, written consent, approved by the West Virginia University Institutional Review Board for the Protection of Human Subjects was obtained. Participants wore snug fitting shorts and shirt for the testing session. Each participant completed a pain assessment survey based on the validated Visual Analog Scale (VAS) Foot and Ankle scale [23]. The VAS survey contains questions about localized pain in the low back, hip/buttocks, upper leg, knee, lower leg, and foot/ankle regions assess occurrence and degree of pain bilaterally during gait, rest, physical activity, and activities of daily living. A score of "0" indicates no pain, and "10" indicated worst possible pain. VAS in each location was calculated as an average of score over all categories in that location. Pregnant subjects who reported pain >3 in any location were placed into the "pregnant pain" group (n = 5). Pregnant participants with no pain (VAS <3) at all locations were categorized in the "pregnant no pain" group (n = 5). Lastly, healthy non-pregnant participants comprised a control group (n = 9). To assess pelvic, lower extremity, and foot alignment, a series of biomechanical measurements using

**Table 1. Demographics for pregnant and control groups.**

|  | Control | Pregnant | p-value |
|---|---|---|---|
| **Age(yrs)** | 22.0±1.1 | 29.6±3.0 | 0.07 |
| **Height(cm)** | 162.7±4.6 | 165.7±6.6 | 0.590 |
| **Pre-Pregnancy Mass(kg)*** | 62.4±6.4 | 72.02±15.6 | 0.001 |

* = p≤0.05

**Table 2. Demographics for pregnant pain and pregnant no pain groups.**

|  | Pain | No Pain | p-value |
|---|---|---|---|
| **Age(yrs)** | 29.4±4.2 | 29.8±1.6 | 0.147 |
| **Height(cm)** | 167.1±8.1 | 163.8±4.4 | 0.160 |
| **Current Mass(kgs)** | 88.1±13.8 | 78.6±13.3 | 0.831 |
| **Pre-pregnancy Mass(kgs)** | 75.6±16.1 | 69.45±16.0 | 0.818 |
| **Weeks Pregnant** | 31.0±1.6 | 32.0±2.0 | 0.855 |
| **# of Pregnancies** | 1.8±1.8 | 1.4±0.6 | 0.136 |

metrics previously suggested might be affected with pregnancy [19, 24]. Measurements were made bilaterally, with the left or right side randomly assessed first. Asymmetry was calculated as the absolute difference between the right and left sides for each metric. A flowchart summarizing variables measures and grouping for statistical analysis can be found in Fig 1.

Foot Width (FW) was measured at the widest part of the forefoot using a standard set of calipers with the participant in stance. Arch Index (AI), an indirect assessment of arch height, was measured using the inked footprint technique (Aetrex Worldwide, Teaneck, NJ) [25]. On the inked footprints a line was drawn from the tip of the second toe to the most posterior aspect of the heel. Distance from the anterior portion of the forefoot (not including the toes) to the posterior heel was measured along this line. This distance was divided into thirds and corresponding markers were placed along the line on the footprint. AI was calculated as the area of midfoot / total area of the footprint [25] using the NIH software ImageJ (NIH, Bethesda, MD). Increased AI indicates a lower arch, as more of the midfoot area is in contact with the ground [25]. Rearfoot angle (RFA) and Pelvic Obliquity (PO) were obtained from frontal plane photographs with the participant standing with feet shoulder width apart. RFA was defined as the subtalar joint angle [26] and PO was defined as the angle of the pelvis based on the angle a line connecting the left and right ASIS with the horizontal. These angles were measured using Image-J software.

Next, several foot alignment measures were obtained using the Arch Height Index Measurement System (JAK Tool, Cranbury, NJ) [27]. Using a set of sliding calipers elevated on two wooden blocks to leave the medial longitudinal arch unsupported, three measurements of each foot were taken: Foot length (FL), truncated FL (the distance from the most posterior aspect of the heel to the head of the first metatarsal), and foot height (the height of the foot at one half of foot length). Measurements were first taken with the participant seated. A goniometer was used to position the ankle such that the line between the first metatarsal head and the lateral malleolus formed a 120˚ angle with the line from the lateral malleolus to the fibular head. The

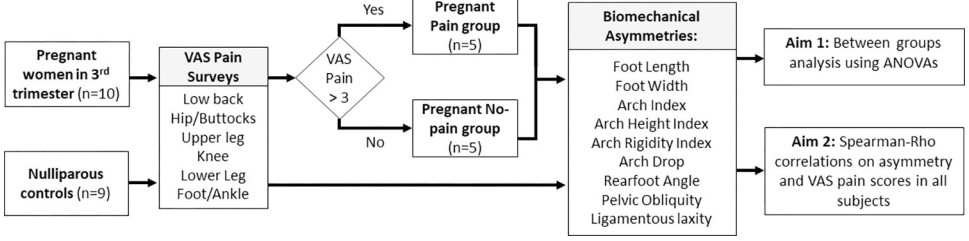

**Fig 1. Data collection and analysis flowchart.** All participants began by completing the VAS pain surveys. Pregnant women were grouped into 'pain' or 'no pain' groups based on these scores with a pain threshold of 3. Biomechanical Asymmetries were quantified for 9 variables and used in statistical analysis.

same measures were taken with the participant standing with weight evenly distributed on both feet. These measurements were used to calculate Arch Height Index (AHI), Arch Rigidity Index (ARI), and Arch Drop (AD). AHI = foot height/truncated FL, where higher values of AHI indicate higher arches [27]. AHI measurements were performed two times. Once with the participants sitting and again while standing. The sitting AHI was only used to quantify Arch rigidity index (ARI) and arch drop (AD), which are measures of arch flexibility. ARI = standing AHI/seated AHI. An ARI of 1 indicates a perfectly rigid arch, while values closer to 0 indicate a more flexible arch. AD = seated foot height—standing foot height.

To assess overall joint laxity, a modified version of the Beighton's Ligamentous Laxity scale was employed [28]. This scale involves a series of five tests for passive flexibility: hyperextension of each knee, touching each thumb to the forearm, extension of each small finger beyond 90˚ and hyperextension of the elbows. We excluded the sixth test from the Beighton Ligamentous Laxity scale, as flexion at the hip to put hands flat on the floor proves difficult in pregnant females due to their increased abdominal volume [24]. Participants were given 1 point for each for each task they were capable for performing for a total of five possible points. The participant was not forced to stretch beyond her comfort level. The higher the score received, the higher the degree of laxity.

### Statistics

Statistical analysis was performed using IBM SPSS Statistics software (Armonk, New York). Demographics (e.g. age, height, weight) of the population were determined. Descriptive statistics, including means and standard deviations, were calculated for continuous data.

The first aim was to examine if differences existed with regard to foot asymmetries in females with pain during pregnancy compared to a group of pregnant females without pain and a control group of never pregnant females. Dependent variables included average bilateral differences for FL, FW, AI, AHI, ARI, AD, RA PO, and Beighton's ligamentous laxity test. Asymmetry measures were defined as the absolute difference between sides, such that all measures were positive, then each measure of asymmetry was tested for normality using the Shapiro-Wilk test. The independent variable was the group (control, pregnant pain, pregnant no pain). An ANOVA was performed on each dependent variable to examine the difference between groups. Tukey post-hoc analysis was performed when appropriate ($\alpha = 0.05$). 95% confidence intervals were calculated for each group and each variable. Eta Squared values were calculated with each ANOVA in order to determine effect size. Effect sizes were considered small between 0.01 and 0.06, medium if between 0.06 and 0.14 and large if greater than 0.14.

The second aim was to examine the relationship of self-reported measures of foot, posterior pelvic, and lumbar spine pain and biomechanical measures of alignment in all of the subjects. For this aim, we used correlation analysis to assess relationships between asymmetry and level of pain ($\alpha = 0.05$). Because these data were not normality distributed, Spearman-Rho nonparametric regressions were performed for each of the nine asymmetry measures. These included (FL, FW, AI, AHI, ARI, AD, RA, PO, and PI), with pelvic pain, low back pain, leg pain, foot pain, and overall general pain.

### Results

The first aim was to quantify foot asymmetries in females with pain during pregnancy compared to pregnant females without pain and a control group of never pregnant females. None of the variables reached statistical significance (>0.05). However, interesting trends towards group differences were observed for mean asymmetries in FW, AI and ARI as calculated by effect size (Fig 2). Effect sizes were found to be large for FW, AI, and ARI ($\eta^2 = 0.17$, 0.20, and

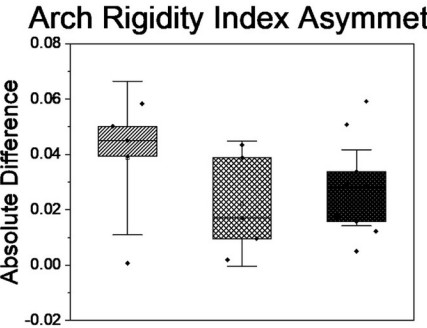

**Fig 2. Box plots for variables with large effect sizes (FW, AI, and ARI asymmetries).** Striped boxed represent absolute differences in each metric for pregnant females reporting no pain, check boxes represent pregnant females with pain, and the dark boxes represent nulliparous controls. Whiskers represent 95% CI for each group.

0.14 respectively). Confidence intervals are presented in the box and whisker plots. Effect sizes and p-values for all variables are presented in Table 3.

The second aim was to examine the relationship of self-reported measures of foot, posterior pelvic, and lumbar spine pain and biomechanical measures of alignment in all of the subjects. Spearman-Rho correlation coefficients revealed significant negative correlations between arch index asymmetry and low back pain (p = 0.005), foot length asymmetry and lower leg pain (p = 0.008), and pelvic obliquity and lower leg pain (p = 0.020). Significant positive correlations were found between foot width asymmetry and knee pain (p = 0.028), as well as arch drop asymmetry and upper leg pain (p = 0.024), knee pain (p = 0.005), and lower leg pain (p = 0.019). These results can be seen in Tables 4 and 5.

## Discussion

This study aimed to determine if asymmetric lower extremity and foot alignment between pregnant females with lower back or lower extremity pain, those not in pain, and nulliparous controls. We also aimed to determine the relationship between asymmetry and severity of pain reported by females reporting pain at the hip, knee foot, or low back during pregnancy. In contrast to our hypothesis that pregnant females would have greater asymmetries and laxity scores compared to control and no pain groups, we were not able to detect any between group differences. However, analysis of effect size revealed interesting trends that agreed with our expected outcomes. In support of our second hypothesis, this study identified relationships between pelvis and foot alignment asymmetric and localized pain severity at the foot, posterior pelvis, and lumbar spine.

**Table 3. Lower extremity asymmetries for each group with effect sizes and p-values.**

| Variable | Pregnant No Pain | Pregnant Pain | Control | Effect Size ($\mu^2$) | p-value |
|---|---|---|---|---|---|
| FL (cm) | 0.1 ± 0.1 | 0.22 ± 0.30 | 0.17 ± 0.12 | 0.032 | 0.781 |
| FW (cm) | 0.1 ± 0.122 | 0.28 ± 0.15 | 0.16 ± 0.16 | 0.174 | 0.24 |
| AI | 0.028 ± 0.007 | 0.015 ± 0.008 | 0.030 ± 0.005 | 0.200 | 0.301 |
| ARI | 0.0.39 ± 0.022 | 0.022 ± 0.018 | 0.028 ± 0.017 | 0.143 | 0.3 |
| AHI | 0.012 ± 0.007 | 0.008 ± 0.007 | 0.005 ± 0.003 | 0.00 | 0.062 |
| AD (cm) | 0.14 ± 0.26 | 0.18 ± 0.08 | 0.11 ± 0.21 | 0.023 | 0.841 |
| RA (deg) | 2.23 ± 1.23 | 2.45 ± 1.38 | 2.26 ± 1.51 | 0.007 | 0.951 |
| PO (deg) | 2.54 ± 1.52 | 2.56 ± 1.88 | 2.14 ± 1.20 | 0.017 | 0.839 |

**Table 4. Spearman-Rho correlation coefficients for low back, hip/buttocks, and upper leg asymmetries +and pain.**

|  | Low Back | | Hip/Buttocks | | Upper Leg | |
|---|---|---|---|---|---|---|
|  | R | p-value | R | p-value | R | p-value |
| FL | 0.127 | 0.61 | 0.085 | 0.729 | -0.271 | 0.262 |
| FW | 0.370 | 0.119 | 0.402 | 0.088 | -0.104 | 0.672 |
| **AI** | **-0.617** | **0.005**** | -0.390 | 0.099 | 0.017 | 0.945 |
| AHI | -0.007 | 0.977 | 0.216 | 0.374 | -0.028 | 0.911 |
| ARI | -0.141 | 0.566 | -0.190 | 0.436 | 0.247 | 0.308 |
| **AD** | 0.280 | 0.246 | 0.097 | 0.693 | **0.514** | **0.024*** |
| RFA | -0.077 | 0.753 | 0.186 | 0.445 | 0.027 | 0.914 |
| PO | -0.071 | 0.778 | -0.375 | 0.125 | -0.432 | 0.073 |

* = p≤0.05

** = p<0.01

In this study, we found no significant difference in FW asymmetries between groups; however, sample size was small and calculating effect sizes provided some insight into trends. There was a large effect size ($\eta^2 = 0.17$) in which pregnant females who reported pain trended towards having greater FW asymmetries than those without pain and controls. Previous examinations of foot widening as a result of pregnancy are conflicting as Ponnapula and Boberg [5] and Wetz et al. [29] both reported increased FW, while Harrison [18] reported no increase in foot width following pregnancy. Harrison et al. [18] had a small sample size (n = 15), and Ponnapula and Boberg [5] relied on anecdotal self-report from participants. Further, Harrison et al. [18] pooled foot measurements from the right and left feet; thus the presence of asymmetries could have washed out any detectable change in foot width.

We also detected large effect size for AI ($\eta^2 = 0.20$) in which pregnant females in pain trended towards having lower AI asymmetry than the other two groups. Our participants were in their third trimester. During this trimester, swelling of the lower leg and foot often occur [16]. If our subjects experienced this edema, their larger foot volume may have hidden asymmetry in arch index [16, 30]. Lastly, ARI trended towards a higher ARI in the pregnant pain group compared to other groups with a large effect size ($\eta^2 = 0.143$). In contrast to this, Segal et al. [19] reported a decreased ARI. However, these measures likely excluded swelling as a

**Table 5. Spearman-Rho correlation coefficients for knee, lower leg, and foot/ankle and pain.**

|  | Knee | | Lower Leg | | Foot/Ankle | |
|---|---|---|---|---|---|---|
|  | R | p-value | R | p-value | R | p-value |
| **FL** | -0.003 | 0.991 | **-0.587** | **0.008**** | -0.161 | 0.509 |
| **FW** | **0.504** | **0.028*** | 0.302 | 0.208 | 0.396 | 0.093 |
| AI | -0.321 | 0.181 | -0.006 | 0.982 | -0.171 | 0.484 |
| AHI | -0.384 | 0.150 | -0.022 | 0.928 | 0.273 | 0.257 |
| ARI | 0.291 | 0.226 | 0.313 | 0.191 | 0.182 | 0.457 |
| **AD** | **0.619** | **0.005**** | **0.534** | **0.019*** | 0.402 | 0.088 |
| RFA | 0.323 | 0.177 | 0.243 | 0.316 | -0.081 | 0.742 |
| **PO** | -0.105 | 0.677 | **-0.542** | **0.020*** | -0.122 | 0.629 |

* = p≤0.05

** = p<0.01

confounding factor because they were taken in the first trimester and 19 weeks post-partum [19] where in the present study, participants were in their third trimester during which foot volume is often increased [16]. We suspect that the presence of swelling could cause a "bottoming out" effect on the arch where the arches may appear rigid, when in fact they are just restricted in movement by the increased foot volume.

No differences were detected in FL, AD, and RA asymmetries between groups. Prior studies have reported increases in FL throughout pregnancy [16, 18, 19]; however, the amount of asymmetric change between the left and right sides has not been previously assessed. Lack of asymmetric changes in AD and RA are supported by previous reports of no previous significant change in these measures throughout pregnancy [18]. Harrison et al. [18] also reported no relationship between PO and pregnancy; however, a trend was noted that the control group had increased PO compared to the pregnant group. This trend could not be confirmed by the current study. Means for PO were similar between our three groups.

In this study, no significant relationships were found between any measured asymmetries and severity of pain when location of pain was not considered; however, foot alignment asymmetries, with the exception of AHI and ARI, were correlated with pain in specific areas. These findings are consistent with Harrison [18] in that AD asymmetries are positively correlated with upper leg, knee and lower leg pain. However, Harrison [18] also reported that AD and ARI asymmetries were correlated to foot and ankle pain, and upper leg pain respectively, but the present study was unable to replicate these results. Moreover, we found no significant correlations between ARI asymmetries and pain at any location. Regardless, this may provide useful information for clinical treatment as reducing or preventing asymmetries in these metrics may help mitigate pain in the respective locations. Footwear and exercise interventions should be explored as they act to improve foot posture, strengthen the lower limb, and reduce asymmetries.

Asymmetries in AI were negatively correlated to low back pain. This makes sense considering that the pregnant females with pain in this study trended towards the least amount of AI asymmetry in the previous analysis. Harrison [18] reported that change in AI throughout the course of a pregnancy was related to pain. While this did not include a metric of asymmetry, it suggests that the pregnant women experiencing may have lower arches [18]. This could lead to an effect of "bottoming out" where less asymmetry exists due to arches being as low as they can go. It is possible that swelling or increased foot volume could produce a similar confounding effect [30].

## Limitations

While the relationships established in this study between alignment asymmetries and site-specific pain may ultimately be useful in the clinical setting, this study is not without limitations. Because we performed these measurements only during pregnancy, it is not clear if the trend towards pregnant females having larger asymmetries in these metrics could be due to asymmetric changes, asymmetric swelling, or previous asymmetry in width such as the asymmetry that existed in the control group. Future work should obtain measurements both before and after pregnancy and measure foot volume to assess swelling and help eliminate some of these confounders.

Other limitations are that pain levels were self-reported and pain threshold can be increased during pregnancy [31, 32] which could impact ability to detect correlations based on pain severity, and our sample size was small. Despite this we were able to find correlations between knee, lower leg, and low back pain and AD, FL, and AI asymmetries respectively. In our study, most of the pain group reported some degree of pain at *every* location, but it was only clinically

significant for n = 1 at the knee, n = 1 and the foot and ankle, and n = 4 at the hip. However, every pregnant female in this study with clinically significant (VAS>3) pain in any location also reported having clinically significant low back pain (VAS = 6.68±1.01) as well as pain in at least one other area that we assessed. Furthermore, no pregnant females reported VAS = 0 in the low back pain category. This warrants further investigation into how these biomechanical factors may be related to low back pain in pregnancy, including individuals who are not experiencing any level of low back pain. Given previous reports that greater degree of pain may be related to pain post-partum [20], it is important to understand how interventions affecting foot posture can prevent or decrease severity of pain. Probing how these asymmetries are related in nulliparous controls and post-partum females will provide important insight to understand lasting relationships to pain. Treating for these asymmetries could be useful to determine if the treatments can alleviate pain, and of particular importance, low back pain.

## Conclusion

Low back and lower extremity pain is extremely prevalent during pregnancy. This study revealed that asymmetric foot posture and pelvic obliquity are related to self-reported lower extremity pain in specific locations. These results help identify areas for clinicians to target and treat pain but may require a larger sample size to fully detect differences in alignment that may exist between pregnant females with and without pain in specific locations. Future work should attempt to uncover if these asymmetries persist or are related to pain post-partum and if treatment during pregnancy can be preventative of pain post-partum.

## Supporting information

**S1 Data.**
(XLSX)

## Author Contributions

**Conceptualization:** Erica M. Casto, Corrie Mancinelli, Petronela Meszaros, Jean L. McCrory.

**Data curation:** Jean L. McCrory.

**Formal analysis:** Erica M. Casto, Jean L. McCrory.

**Investigation:** Erica M. Casto, Corrie Mancinelli, Petronela Meszaros, Jean L. McCrory.

**Methodology:** Erica M. Casto, Corrie Mancinelli, Jean L. McCrory.

**Project administration:** Erica M. Casto.

**Supervision:** Jean L. McCrory.

**Writing – original draft:** Erica M. Casto, Jean L. McCrory.

**Writing – review & editing:** Corrie Mancinelli, Petronela Meszaros.

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
