## [Decision Letter · Decision Letter 0]

2 Nov 2022

PONE-D-22-28124Asymmetric changes in foot anthropometry with pregnancy may be related to onset of lower limb and low back painPLOS ONE

Dear Dr. McCrory,

Thank you for submitting your manuscript to PLOS ONE. After careful consideration, we feel that it has merit but does not fully meet PLOS ONE’s publication criteria as it currently stands. Therefore, we invite you to submit a revised version of the manuscript that addresses the points raised during the review process.

We look forward to receiving your revised manuscript.

Kind regards,

Yaodong Gu

Academic Editor

PLOS ONE

Journal Requirements:

Additional Editor Comments:

Please check the questions raised by the reviewers.

Reviewers' comments:

Reviewer's Responses to Questions

**Comments to the Author**

1. Is the manuscript technically sound, and do the data support the conclusions?

Reviewer #1: Yes

Reviewer #2: Yes

2. Has the statistical analysis been performed appropriately and rigorously? 

Reviewer #1: Yes

Reviewer #2: Yes

3. Have the authors made all data underlying the findings in their manuscript fully available?

Reviewer #1: Yes

Reviewer #2: Yes

4. Is the manuscript presented in an intelligible fashion and written in standard English?

Reviewer #1: Yes

Reviewer #2: Yes

5. Review Comments to the Author

Reviewer #1: This manuscript is titled “Asymmetric changes in foot anthropometry with pregnancy may be related to onset of the lower limb and low back pain”. There is a great insight into investigating foot asymmetry and pain in pregnancy. However, you mentioned in the abstract section that the purpose of this article was to compare asymmetric alignment in females with and without pain during pregnancy and to assess the relationship between asymmetric alignment and pain severity, and in the results section, you have mentioned that the “first aim” and the “second aim”, we did not understand what you mean by your purpose.

We can’t quite understand the purpose of this manuscript. Does the author want to investigate the symmetric alignment in females with and without pain during pregnancy and to assess the relationship between asymmetric alignment and pain severity or does the author want to quantify foot asymmetries in females with pain during pregnancy compared to pregnant females without pain and a control group of never pregnant females? These two purposes are different, but important for this research, and we suggested the author revise them clearly in the article.

Specific comments are shown below:

We consider the results section of this manuscript mentioned that pain during pregnancy can be predictive of pain post-partum, how to conclude this viewpoint? In this research or previous report? Please provide relevant reference documentation.

We did not understand the result that pain during pregnancy can be predictive of pain post-partum, the participants in this manuscript concluded ten pregnant females in their third trimester and nine nulliparous controls, how to get the relationship between these two groups?

The sample size used in this article is small. Is its between-group analysis statistically significant? Please explain this clearly to the author. Why such an analysis and comparison if it doesn't make much sense?

At the same time, the sample size of the subjects mentioned by the author in the limitations, which is also the concern of the reviewer

Reviewer #2: Review comment

This manuscript entitled “Asymmetric changes in foot anthropometry with pregnancy may be related to onset of lower limb and low back pain” primarily aimed to investigate the relationship between asymmetric in lower extremity alignment or foot posture and pain severity. The results of this study provide guidance for public health and bioengineering. While it is a very interesting topic. But I think this manuscript has some flaws to fill in before it can be published in a journal. There are several questions should be addressed, which list below. I give a major revision for this manuscript.

Specific comments

1. In the abstract part. In the opinion of reviewer, the author provided too much background descriptions in this part, which may be too long-winded. I suggest that the authors provide more detailed descriptions of the results and conclusions of this study in this part.

2. “Ten pregnant females in their third trimester and nine nulliparous controls participated. Informed consent was obtained.” The authors recruited only 10 pregnant females in their trimester and none nulliparous contrils participated. How was this sample size calculated?(Line 43-44)

3. In the Introduction part. “Approximately 50% of pregnant females report low back and lower extremity pain during pregnancy, with nearly a quarter reporting severe pain reducing overall quality of life” While the Ostgaard 1991 and Vullo 1996 references are seminal, there are more recent publications that discussed the low back and lower extremity pain of pregnant females. Please provide the latest references in this sentence. (Line 63-64)

4. “thus the purpose of this study was to assess asymmetries between left and right limbs in foot length (FL), foot width (FW), arch index (AI), arch rigidity index (ARI), arch height index (AHI), arch drop (AD), rearfoot angle (RFA), and pelvic obliquity (PO), as well as overall joint laxity during pregnancy.” On what basis did the authors choose these variables. (Line 81-83)

5. In the reviewer’s opinion, the authors’ description of bilateral asymmetry is not complete enough, I suggest that the authors add more descriptions about the biomechanical asymmetry in pregnant famale.

6. In the Methods part. “Nineteen healthy females ages 21-34 participated in this study.” How was this sample size calculated? (Line 99-100)

7. The reviewer recommended that the authors provide one or more citations to support the recruitment and screening criteria of participants in this study. (Line 99-100)

8. In the reviewer's opinion, the authors provide a lot of detailed procedures and stutistics, which makes the description too lengthy. To make the process clearer to the reader, it is recommended to add a flowchart. (Line 100-105)

9. In the results part, in the opinion of the reviewer, the pixel in Figure 1 is too blurred, please replace a clearer figure.

10. In the discussion part, “Because arch rigidity is based on the flexibility of the foot, it is important to note that these measures could also be affected by swelling.” Please add a reference to support this sentence. (Line 255-257)

11. What are the limitations of this study? Please provide relevant description.

12. In the Conclusion part. In the opinion of the reviewer, the description in the conclusion part was too verbose, and the reviewer suggests that the authors should abbreviate the section and focus on the main findings of this study.

6. PLOS authors have the option to publish the peer review history of their article (what does this mean?). If published, this will include your full peer review and any attached files.

Reviewer #1: No

Reviewer #2: No

---

## [Author Response · Author response to Decision Letter 0]

13 Apr 2023

We are thankful to the reviewers for their detailed read-through of our manuscript and their insightful input. We appreciate their comments and have done our best to address them. 

Reviewer 1:

Reviewer #1: This manuscript is titled “Asymmetric changes in foot anthropometry with pregnancy may be related to onset of the lower limb and low back pain”. There is a great insight into investigating foot asymmetry and pain in pregnancy. 

However, you mentioned in the abstract section that the purpose of this article was to compare asymmetric alignment in females with and without pain during pregnancy and to assess the relationship between asymmetric alignment and pain severity, and in the results section, you have mentioned that the “first aim” and the “second aim”, we did not understand what you mean by your purpose. We can’t quite understand the purpose of this manuscript. Does the author want to investigate the symmetric alignment in females with and without pain during pregnancy and to assess the relationship between asymmetric alignment and pain severity or does the author want to quantify foot asymmetries in females with pain during pregnancy compared to pregnant females without pain and a control group of never pregnant females? These two purposes are different, but important for this research, and we suggested the author revise them clearly in the article.

Thank you for noticing that the purpose stated in the abstract of our original submission did not align with what was done in the study. Therefore, we amended the purpose in the abstract to read as follows: “The purpose of this study was twofold: 1) to compare asymmetric alignment in pregnant females with and without pain during pregnancy and in nulliparous controls and 2) to assess the relationship between asymmetric alignment and pain severity in all participants.”

We consider the results section of this manuscript mentioned that pain during pregnancy can be predictive of pain post-partum, how to conclude this viewpoint? In this research or previous report? Please provide relevant reference documentation.

We have added reference that state the pain in pregnancy is predictive of pain post-partum in the final paragraph of the discussion on line 322.

We did not understand the result that pain during pregnancy can be predictive of pain post-partum, the participants in this manuscript concluded ten pregnant females in their third trimester and nine nulliparous controls, how to get the relationship between these two groups?

We agree that our study cannot report this because we did not assess pain post-partum. Rather, our purpose was to reiterate the relevance of our study in relating pain to malalignment. If pain during pregnancy was short term and resolved shortly after delivery, this pain would be of little concern. However, the pain is a long-term issue for many women. Our purpose was to examine alignment and pain and this is important because this pain could be predictive of pain post-partum, and we now cite the statements about this relationship. 

The sample size used in this article is small. Is its between-group analysis statistically significant? Please explain this clearly to the author. Why such an analysis and comparison if it doesn't make much sense?

On line 214, we state that none of the between-group analyses found statistical significance for any variable tested. 

At the same time, the sample size of the subjects mentioned by the author in the limitations, which is also the concern of the reviewer

We acknowledge that our sample size is small. We actively searched for participants for several months. We live in a small rural college town and our sample size was limited by participant availability and willingness to participate. In the end, we needed to complete this study in order for the first author on the study to complete her degree requirements and graduate from our MS program. We did not continue the study because we did not want a second investigator taking the alignment measurements due to inter-tester variability. 

Reviewer #2:

This manuscript entitled “Asymmetric changes in foot anthropometry with pregnancy may be related to onset of lower limb and low back pain” primarily aimed to investigate the relationship between asymmetric in lower extremity alignment or foot posture and pain severity. The results of this study provide guidance for public health and bioengineering. While it is a very interesting topic. But I think this manuscript has some flaws to fill in before it can be published in a journal. There are several questions should be addressed, which list below. I give a major revision for this manuscript.

Specific comments

In the abstract part. In the opinion of reviewer, the author provided too much background descriptions in this part, which may be too long-winded. I suggest that the authors provide more detailed descriptions of the results and conclusions of this study in this part.

We shortened the detailed of the methods and added the following sentence to the results section of the abstract: “No statistical differences (p>0.05) were found between pregnant females with or without pain and controls for any of the metrics.” 

“Ten pregnant females in their third trimester and nine nulliparous controls participated. Informed consent was obtained.” The authors recruited only 10 pregnant females in their trimester and none nulliparous contrils participated. How was this sample size calculated?(Line 43-44)

This was a sample size of convenience. We had hoped to be able to recruit more participants but were unable to find pregnant women in their third trimester who were willing to come in to for testing. We had to close enrollment for this study in order for the first author to complete her degree requirements and graduate from our MS program. We did not continue the study because we did not want a second investigator taking the alignment measurements because no one else in the lab group was trained to measure these measurements yet. 

In the Introduction part. “Approximately 50% of pregnant females report low back and lower extremity pain during pregnancy, with nearly a quarter reporting severe pain reducing overall quality of life” While the Ostgaard 1991 and Vullo 1996 references are seminal, there are more recent publications that discussed the low back and lower extremity pain of pregnant females. Please provide the latest references in this sentence. (Line 63-64)

We have added more recent references: Kesikburun et al 2018; Gutke et al 2017; Weis et al 2018; and more recent reference for post-partum Tavares et al 2020 

“thus the purpose of this study was to assess asymmetries between left and right limbs in foot length (FL), foot width (FW), arch index (AI), arch rigidity index (ARI), arch height index (AHI), arch drop (AD), rearfoot angle (RFA), and pelvic obliquity (PO), as well as overall joint laxity during pregnancy.” On what basis did the authors choose these variables. (Line 81-83)

We selected these variables because 1) they have been reported previously in the literature as being related to body alignment and/or pain and 2) we had the capability to assess them in our laboratory. These have been reported by prior authors cited in our introduction, primarily Damen (20) and Harrison (18).

In the reviewer’s opinion, the authors’ description of bilateral asymmetry is not complete enough, I suggest that the authors add more descriptions about the biomechanical asymmetry in pregnant female.

To address this concern, we specifically added that our “asymmetry” is defined as the absolute difference between the right and left sides on lines 87 and 135.

In the Methods part. “Nineteen healthy females ages 21-34 participated in this study.” How was this sample size calculated? (Line 99-100)

Please see our earlier comment on this. This was a sample size of convenience. However, even though the sample size was small, we did find statistical significance in our 2nd aim: to quantify relationships between asymmetry and pain. 

The reviewer recommended that the authors provide one or more citations to support the recruitment and screening criteria of participants in this study. (Line 99-100)

We have added references to our exclusion criteria to support their validity in our study. We selected these criteria due to their effect on pain e.g. scoliosis and injury may increase self-reported pain, but diabetes and other forms of neuropathy may decease self-reported pain.

In the reviewer's opinion, the authors provide a lot of detailed procedures and stutistics, which makes the description too lengthy. To make the process clearer to the reader, it is recommended to add a flowchart. (Line 100-105) 

Thank you for this suggestion. We have included a flow chart as figure 1. 

9. In the results part, in the opinion of the reviewer, the pixel in Figure 1 is too blurred, please replace a clearer figure. 

 We have replaced this with a higher quality image.

10. In the discussion part, “Because arch rigidity is based on the flexibility of the foot, it is important to note that these measures could also be affected by swelling.” Please add a reference to support this sentence. (Line 255-257)

We have rewritten this paragraph and this sentence no longer exists in its same form. However, the concept is still in that paragraph. We cite Alvarez et al (16) for the concept that swelling may affect arch due to a ‘bottoming out’ of the arch due to edema.

11. What are the limitations of this study? Please provide relevant description.

In our prior submission, the study limitations were interspersed throughout the discussion. However, we agree with your suggestion to create a separate paragraph to address specific limitations. Therefore, we have reworked the discussion and now include a limitations section. 

12. In the Conclusion part. In the opinion of the reviewer, the description in the conclusion part was too verbose, and the reviewer suggests that the authors should abbreviate the section and focus on the main findings of this study.

We have reworked this section to omit material that may not be as relevant here and focus on the main findings.

---

## [Decision Letter · Decision Letter 1]

27 Jun 2023

PONE-D-22-28124R1Asymmetric changes in foot anthropometry with pregnancy may be related to onset of lower limb and low back painPLOS ONE

Dear Dr. McCrory,

Thank you for submitting your manuscript to PLOS ONE. After careful consideration, we feel that it has merit but does not fully meet PLOS ONE’s publication criteria as it currently stands. Therefore, we invite you to submit a revised version of the manuscript that addresses the points raised during the review process.

We look forward to receiving your revised manuscript.

Kind regards,

Yaodong Gu

Academic Editor

PLOS ONE

**Additional Editor Comments:**

The authors shall provide enough data to answer the limited number of subjects.

Reviewers' comments:

Reviewer's Responses to Questions

**Comments to the Author**

1. If the authors have adequately addressed your comments raised in a previous round of review and you feel that this manuscript is now acceptable for publication, you may indicate that here to bypass the “Comments to the Author” section, enter your conflict of interest statement in the “Confidential to Editor” section, and submit your "Accept" recommendation.

Reviewer #1: (No Response)

Reviewer #2: (No Response)

2. Is the manuscript technically sound, and do the data support the conclusions?

Reviewer #1: (No Response)

Reviewer #2: (No Response)

3. Has the statistical analysis been performed appropriately and rigorously? 

Reviewer #1: (No Response)

Reviewer #2: (No Response)

4. Have the authors made all data underlying the findings in their manuscript fully available?

Reviewer #1: (No Response)

Reviewer #2: (No Response)

5. Is the manuscript presented in an intelligible fashion and written in standard English?

Reviewer #1: (No Response)

Reviewer #2: (No Response)

6. Review Comments to the Author

Reviewer #1: This manuscript entitled “Asymmetric changes in foot anthropometry with pregnancy may be related to onset of lower limb and low back pain” primarily aimed to compare asymmetric alignment in females with and without pain during pregnancy and to assess the relationship between asymmetric alignment and pain severity. To enhance the quality of the manuscript, revise suggestions are given below.

There are a number of problems with this article that cannot be fixed.

First off, the artificial measurement of foot characteristics has a significant inaccuracy. Using a foot scanner could be more objective in this case, in my opinion. Additionally, the VAS pain is only suitable for supplemental reference. If conclusions are drawn using this data as the benchmark, it is insufficient.

Second, the sample size used in this research is too little to adequately support the results. Only 5 women who were pregnant in all experienced this kind of agony. And this topic should be treated differently for pregnant women. Pregnancy timing is also significant. This was not considered by the writers.

Last but not least, if the author intends to assess the relationship between bone discomfort in pregnant women and foot form characteristics. I believe this research needs further validation.

Reviewer #2: Thanks to the authors for their valuable contributions, I think the manuscript has met the criteria for publication.

7. PLOS authors have the option to publish the peer review history of their article (what does this mean?). If published, this will include your full peer review and any attached files.

Reviewer #1: No

Reviewer #2: No

---

## [Author Response · Author response to Decision Letter 1]

21 Jul 2023

We would like to thank the Reviewers and the Editor for their input into this manuscript. The Editor asked us to provide enough data to answer the limited number of subjects. Reviewer 1 listed several issues that render the study, in their opinion, unpublishable. We have provided rebuttals to these issues below. Editor and Reviewer comments are shown in regular black font while our rebuttals are shown indented in blue font. 

Editor Comment:

The authors shall provide enough data to answer the limited number of subjects.

We agree that our small sample size is a major limitation of this study. However, we feel that our study has merit and provides valuable information to clinicians and others who study pain, or the prevention of pain, in pregnancy. This was mentioned on line 311 where we mention sample size being small is a limitation.

We addressed that the significant results may be limited by sample size, referring to the correlations in our Specific Aim 2. With regard to Specific Aim 1 in which we compared groups using an ANOVA, nothing was significant. We mention this in the results and then we discuss effect sizes in the discussion which are independent of sample size. We make no conclusions about those results - only the correlations (Specific Aim 2). On lines 250-251, we added the statement: “In this study, we found no significant differences in FW asymmetries between groups; however, sample size was small and calculating effect sizes provided some insight into trends. There was a large effect size (η2=0.17) in which pregnant females who reported pain trended towards having greater FW asymmetries than those without pain and controls.”

Reviewer 1’s Comments:

First off, the artificial measurement of foot characteristics has a significant inaccuracy. Using a foot scanner could be more objective in this case, in my opinion. 

We acknowledge that medical imaging may provide more accurate data than the Arch Height Index and Arch Index. However, we believe that these measures are useful tools in their own right with sufficient validity for the current study. 

The Arch Height Index and Arch Index are widely used measurements in biomechanics research. A search of GoogleScholar with the keywords “Arch Height Index” reveals 988 publications which include this measurement. In specifically looking at pregnancy, a GoogleScholar search of “Arch Height Index” and “pregnancy” found 71 publications on the topic. Similarly, a search of GoogleScholar with the keywords “Arch Index” results in 4,040 publications using this measure. If “Pregnancy” is added as a search term, then 212 publications are listed as having Arch Index and pregnancy in them. Given that so many peer-reviewed publications have used these tools, we assert that the biomechanics scientific community accepts these methods as valid. 

Arch Height Index: Butler et al. (2008) established the reliability (ICC) of the AHI to be 0.94. GoogleScholar metrics report that this reference article has been cited by 179 other publications. The Weimar and Shroyer (2013) publication which reports normative AHI values for young women has been cited in 29 publications. 

Arch Index: The seminal publication defining Arch Index was by Cavanagh and Rodgers (1987). This work has been cited in 949 subsequent publications that have used this measure. McCrory et al. (1997) reported a correlation of r=0.67 between footprint obtained Arch Index and true navicular height obtained via a radiograph. Chu et al. (1995) reported a correlation of r=0.70 between arch index obtained via a digital image and that obtained via footprint. 

We acknowledge that lower extremity edema associated with pregnancy will have a negative effect on the accuracy of Arch Height Index and Arch Index. However, we included this as a limitation of the study in our manuscript. 

References:

Butler RJ, Hillstrom H, Song J, Richards CJ, and Davis IS. Arch Height Index Measurement System: Establishment of Reliability and Normative Values. Journal of the American Podiatric Medical Association. 98(2): 102-106, 2008. 

Cavanagh PR and Rodgers MM. The arch index: A useful measure from footprints. Journal of Biomechanics. 20(5): 547-551, 1987.

Chu WC, Lee SH, Chu W, Wang TJ, Lee MC. The use of arch index to characterize arch height: a digital image processing approach. IEEE Transactions on Biomedical Engineering. 42(11): 1088-1093, 1995.

McCrory JL, Young MJ, Boulton AJM, and Cavanagh PR. Arch index as a predictor of arch height. The Foot. 7(2): 79-81, 1997.

Weimar WH and Shroyer JF. Arch Height Index Normative Values of College-Aged Women Using the Arch Height Index Measurement System. Journal of the American Podiatric Medical Association. 103(3): 213-217, 2013. 

Additionally, the VAS pain is only suitable for supplemental reference. If conclusions are drawn using this data as the benchmark, it is insufficient.

Hunt and Hurwit (2013) did a systematic review on all clinical articles reporting on foot and ankle pain in six orthopedic journals over a ten-year period. They concluded that the American Orthopaedic Foot and Ankle Society Scale (AOFAS) was the most common measure used for pain, as it was included 55.9% of the 878 publications they reviewed. The VAS scale was the second most commonly used pain scale, as it was used in 22.9% of the studies. The VAS Foot and Ankle differs from the AOFAS score in that the VAS is a survey of perceived pain and function, whereas the AOFAS score requires the investigator to perform a physical examination of the subject. Other scales of pain and function include the Ankle Osteoarthritis Score (AOS) and the Foot Function Index (FFI). The AOS scale of lower extremity pain is only validated in patients with ankle OA and would not be valid for use in our study (Domsic and Saltzman, 1998). Similarly, the FFI is validated for low functional individuals with foot disorders or patients with arthritis or foot and ankle problems and is not considered appropriate for higher-functioning individuals (Agel et al, 2005). The VAS Foot and Ankle scale that we used in this study was validated for use in a variety of pathological conditions (Stuber et al., 2011). 

The Visual Analog Scale (VAS) is frequently employed in biomechanics research. A search of GoogleScholar for “visual analog scale” and “biomechanics” yielded more than 24,300 biomechanics-related publications that use the VAS as a measure of subject perception of pain. A GoogleScholar search using the keywords “VAS Foot and Ankle” and “biomechanics” resulted in 37 biomechanics studies that used the VAS Foot and Ankle to evaluate subject pain. 

Given the number of studies who have also used the VAS Foot and Ankle and the fact that it is a survey-based scale of perceived pain and is not based on a physical assessment, we determined that the VAS Foot and Ankle was the best scale to use for our study.

Kelly (2001) determined that clinically-significant patient-reported pain improvement or worsening corresponded to a change of ~1.2 on the VAS scale across three groups of subjects who reported “mild”, “moderate”, and “severe” pain, respectively. We used a VAS score of 3 as the delineation between our “pregnant pain” and “pregnant no pain” groups, in accordance with Kelly (2001). 

We acknowledge that pain-level is subjective and is based on each subject’s perception of their discomfort. The validity of the VAS for use in pain measurement has been questioned. Landy and Sheppeard (1985) asserted that the results of the VAS are non-linear and prone to bias and thus it should not be used as a serial measure of pain severity. We did not use the VAS as a serial measure. Our participants each made one visit for the study. 

References:

Agel J, Beskin JL, Brage M, et al. Reliability of the foot function index: a report of the AOFAS outcomes committee. Foot and Ankle International. 26: 962-967, 2005.

 Domsic RT and Saltzman CL: Ankle osteoarthritis scale. Foot and Ankle International. 19: 466-471, 1998. 

Hunt KJ and Hurwit DBA. Use of patient-reported outcome measures in foot and ankle research. The Journal of Bone and Joint Surgery. 95(16): e118, 2013. 

Kelly AM. The minimum clinically significant different in visual analogue scale pain score does not differ with severity of pain. Emergency Medicine Journal. 2001. http://dx.doi.org/10.1136/emj.18.3.205

Langley GB and Sheppeard H. The visual analogue scale: its use in pain measurement. Rheumatology International. 5: 145-148, 1985. 

Stuber J, Zech S, Bay R, Qazzaz, A, and Richter M. Normative data of the Visual Analogue Scale Foot and Ankle (VAS FA) for pathological conditions. Foot and Ankle Surgery. 17(3), 166-172, 2011). 

Second, the sample size used in this research is too little to adequately support the results. Only 5 women who were pregnant in all experienced this kind of agony. 

Our results were not significant thus we did not discuss them as such. We mentioned trends based on effect sizes and made no conclusions regarding this aim. We simply stated we needed a larger sample size to be able to detect differences in alignment that may exist – line 330 we said “differences in alignment that may exist”.

And this topic should be treated differently for pregnant women. Pregnancy timing is also significant. This was not considered by the writers.

We do not understand this comment. What topic specifically: comparing asymmetry in pregnant women who report pain, those who do not report pain, and non-pregnant controls, or looking at the relationship between asymmetry magnitude and pain level? We do not understand what is meant by “treated differently for pregnant women”. 

We also do not understand what is meant by “pregnancy timing”. All of our pregnant participants were in their third trimester. The subjects in the pregnant pain group were at 31.0 ± 1.6 weeks pregnant while the subjects in the pregnant no-pain group were at 32.0 ± 2.0 weeks pregnant. Thus, our pregnant participants were all within a few weeks of each other. These data are provided in Table 1. 

Last but not least, if the author intends to assess the relationship between bone discomfort in pregnant women and foot form characteristics. I believe this research needs further validation.

We apologize, but we do not understand this question. We do not use the word “bone” or name one of the lower extremity bones (femur, tibia, fibula, calcaneus, etc.) as a site of pain or discomfort in the submitted manuscript. We assessed regional pain in the lower back, hip/buttocks, upper leg, knee, lower leg, and foot/ankle. This pain can be any type of musculoskeletal pain, such as joint pain or soft tissue pain. 

Reviewer 2’s comments:

Thanks to the authors for their valuable contributions, I think the manuscript has met the criteria for publication.

Thank you very much for your kind words and approval of our revisions to our earlier work.

---

## [Decision Letter · Decision Letter 2]

13 Sep 2023

Asymmetric changes in foot anthropometry with pregnancy may be related to onset of lower limb and low back pain

PONE-D-22-28124R2

Dear Dr. McCrory,

We’re pleased to inform you that your manuscript has been judged scientifically suitable for publication and will be formally accepted for publication once it meets all outstanding technical requirements.

Kind regards,

Yaodong Gu

Academic Editor

PLOS ONE

Additional Editor Comments (optional):

The authors have done a well revision, it could be accepted.

Reviewers' comments:

Reviewer's Responses to Questions

**Comments to the Author**

1. If the authors have adequately addressed your comments raised in a previous round of review and you feel that this manuscript is now acceptable for publication, you may indicate that here to bypass the “Comments to the Author” section, enter your conflict of interest statement in the “Confidential to Editor” section, and submit your "Accept" recommendation.

Reviewer #1: (No Response)

Reviewer #3: All comments have been addressed

2. Is the manuscript technically sound, and do the data support the conclusions?

Reviewer #1: Partly

Reviewer #3: Yes

3. Has the statistical analysis been performed appropriately and rigorously? 

Reviewer #1: No

Reviewer #3: Yes

4. Have the authors made all data underlying the findings in their manuscript fully available?

Reviewer #1: Yes

Reviewer #3: Yes

5. Is the manuscript presented in an intelligible fashion and written in standard English?

Reviewer #1: Yes

Reviewer #3: Yes

6. Review Comments to the Author

Reviewer #1: (No Response)

Reviewer #3: Review comment

Thanks to the author for the revision of the manuscript and congratulation for the nice work.

In my point of view, the main limitation of this research is the sample size, but author have mentioned this and explained in the discussion section. The results of this study may be limitation by the sample size, but it’s still could provide valuable information to guide people to consider and related those issue with the asymmetric changes in foot anthropometry. The methods section was reported clearly and result section was fully discussed in the discussion section.

In my point of view, the manuscript has met the criteria for publication.

7. PLOS authors have the option to publish the peer review history of their article (what does this mean?). If published, this will include your full peer review and any attached files.

Reviewer #1: No

Reviewer #3: **Yes: **Yuqi He

---

## [Editor Report · Acceptance letter]

15 Feb 2024

PONE-D-22-28124R2 

PLOS ONE

Dear Dr. McCrory, 

I'm pleased to inform you that your manuscript has been deemed suitable for publication in PLOS ONE. Congratulations! Your manuscript is now being handed over to our production team.

Kind regards, 

on behalf of

Professor Yaodong Gu 

Academic Editor

PLOS ONE